# Improving Zero-shot Generalization in Offline Reinforcement Learning using Generalized Similarity Functions

## Abstract

Reinforcement learning (RL) agents are widely used for solving complex sequential decision making tasks, but still exhibit difficulty in generalizing to scenarios not seen during training. While prior online approaches demonstrated that using additional signals beyond the reward function can lead to better generalization capabilities in RL agents, i.e. using self-supervised learning (SSL), they struggle in the offline RL setting, i.e. learning from a static dataset. We show that performance of online algorithms for generalization in RL can be hindered in the offline setting due to poor estimation of similarity between observations. We propose a new theoretically-motivated framework called Generalized Similarity Functions (GSF), which uses contrastive learning to train an offline RL agent to aggregate observations based on the similarity of their expected future behavior, where we quantify this similarity using *generalized value functions*. We show that GSF is general enough to recover existing SSL objectives while also improving zero-shot generalization performance on a complex offline RL benchmark, offline Procgen.

## 1 Introduction

Reinforcement learning (RL) is a powerful framework for solving complex tasks that require a sequence of decisions. The RL paradigm has allowed for major breakthroughs in various fields, e.g. outperforming humans on video games (Mnih et al., 2015; Schwarzer et al., 2020), controlling stratospheric balloons (Bellemare et al., 2020) and learning reward functions from robot manipulation videos (Chen et al., 2021). More recently, RL agents have been tested in a generalization setting, i.e. in which training involves a finite number of related tasks sampled from some distribution, with a potentially distinct sampling distribution during test time (Cobbe et al., 2019; Song et al., 2019). The main issue for designing generalizable agents is the lack of on-policy data from tasks not seen during training: it is impossible to enumerate all variations of a real-world environment during training and hence the agent must extrapolate from a (limited) training task collection onto a broader set of problems. Since the learning agent is given no training data from test-time tasks, this problem is referred to as zero-shot generalization. In our work, we are interested in the problem of zero-shot generalization where the difference between tasks is predominantly due to perceptually distinct observations. An example of this setting is any environment with distractor features (Cobbe et al., 2020; Stone et al., 2021), i.e. features with no dependence on the reward signal nor the agent's decisions. This generalization setting has recently received much attention (Liu et al., 2020a; Agarwal et al., 2021; Mazoure et al., 2021), due to its particular relevance to real-world scenarios, for example deploying a single autonomous driving agent at day or at night.

Generalization capabilities of an agent can be analyzed through the prism of *representation learning*, under which the agent's current belief about a rich and high-dimensional environment are summarized in a low-dimensional entity, called a representation. Recent work in online RL has shown that learning state representations with specific properties such as disentanglement (Higgins et al., 2017) or linear separability (Lin et al., 2020) can improve zero-shot generalization performance. Achieving this with limited data (i.e. offline RL) is challenging, since the representation will have a large estimation error over regions of low data coverage. A common solution to mitigate this task-specific overfitting and extracting the most information out of the data consists in introducing auxiliary learning signals other than instantaneous reward (Raileanu and Fergus, 2021). As we show later in the

paper, many such signals already contained in the dataset can be used to further improve generalization performance. For instance, the generalization performance of PPO on Procgen remains limited even when training on 200M frames, while generalization-oriented agents (Raileanu and Fergus, 2021; Mazoure et al., 2021) can outperform it by leveraging additional auxiliary signals. However, a major issue with the aforementioned methods is their exorbitant reliance on online access to the environment, an impractical restriction for real-world scenarios.

In contrast, in many real-world scenarios access to the environment is restricted to an offline, fixed dataset of experience (Ernst et al., 2005; Lange et al., 2012). A natural limitation for generalization from offline data is that policy improvement is dependent on dataset quality. Specifically, high-dimensional problems such as control from pixels require large amounts of training experience: a standard training of PPO (Schulman et al., 2017) for 25 million frames on Procgen (Cobbe et al., 2020) generates more than 300 Gb of data, an impractical amount of data to share for offline RL research. Improving zero-shot generalization performance from an offline dataset of high-dimensional observations is therefore a hard problem due to limitations on dataset size and quality.

In this work, we are interested in improving zero-shot generalization across a family of Partially-Observable Markov decision processes (MDPs, Puterman, 1990) in an offline RL setting, i.e. by training agents on a fixed dataset. We hypothesize that in order for an RL agent to be able to generalize across perceptually different POMDPs without adaptation, observations with similar future behavior should be assigned to close representations. We use the generalized value function (GVF) framework (Sutton et al., 2011) to capture future behavior with respect to an arbitrary instantaneous signal (called cumulant) for a given state. The choice of cumulant then determines the nature of the behavioral similarity that is encouraged for generalization. For example, using reward as the signal gives rise to of reward-aware behavioral similarity such as bisimulation (Ferns et al., 2004; Li et al., 2006; Castro, 2020; Zhang et al., 2020); using future state-action counts encourages reward-free behavioral similarity (Misra et al., 2020; Liu et al., 2020a; Agarwal et al., 2021; Mazoure et al., 2021).

Our main contributions are as follows:

1. We propose Generalized Similarity Functions (GSF), a novel self-supervised learning algorithm for reinforcement learning, that aggregates latent representations by the future behavior (or generalized value function) under their respective observations.

2. We devise a new benchmark constructed to test zero-shot generalization of offline RL algorithms: offline Procgen. It consists of 5M transitions from 200 related levels of 16 distinct games.

3. We evaluate performance of GSF and other baseline methods on offline Procgen, and show that GSF outperforms both previous state-of-the-art offline RL and representation learning baselines on the entire distribution of levels.

4. We analyze the theoretical properties of GSF and describe the impact of hyperparameters and cumulant functions on empirical behavior.

## 2 RELATED WORKS

**Generalization in reinforcement learning**   Generalizing a model's predictions across a variety of unseen, high-dimensional inputs has been extensively studied in the static supervised learning setting (Bartlett, 1998; Triantafillou et al., 2019; Valle-Pérez and Louis, 2020; Liu et al., 2020b). Generalization in RL has received a lot of attention: extrapolation to unseen rewards (Barreto et al., 2016; Misra et al., 2020), observations (Zhang et al., 2020; Raileanu and Fergus, 2021; Liu et al., 2020a; Agarwal et al., 2021; Mazoure et al., 2021) and transition dynamics (Ball et al., 2021). Each generalization scenario is best solved by their respective set of methods: sufficient exploration (Misra et al., 2020; Agarwal et al., 2020), auxiliary learning signals (Srinivas et al., 2020; Mazoure et al., 2020; Stooke et al., 2021) or data augmentation (Ball et al., 2021; Sinha and Garg, 2021). Data augmentation is a promising technique, but typically relies on handcrafted domain information, which might not be available *a priori*. In fact, we will show in our experiments that generalization in the offline RL setting is poor even when using such handcrafted data augmentations, without additional representation learning mechanisms. In this work, we posit that representation learning should use instantaneous auxiliary signals in order to prevent overfitting onto a unique signal (e.g. reward

across tasks) and improve generalization performance. Theoretical generalization guarantees have only been provided so far for limited scenarios, mostly for bandits (Swaminathan and Joachims, 2015), linear MDPs (Boyan and Moore, 1995; Wang et al., 2021b; Nachum and Yang, 2021) and across reward functions (Castro and Precup, 2010; Barreto et al., 2016; Wang et al., 2021a; Touati and Ollivier, 2021).

**Representation learning**   For simple POMDPs, near-optimal policies can be found by optimizing for the reward alone. However, more complex settings may require additional auxiliary signals in order to find state abstractions better suited for control. The problem of learning meaningful state representations (or abstractions) for planning and control has been extensively studied previously (Jong and Stone, 2005; Li et al., 2006), but saw real breakthroughs only recently, in particular due to advances in self-supervised learning (SSL). Outside of RL, SSL has achieved spectacular results by closing the gap between unsupervised and supervised learning on certain datasets (Hjelm et al., 2018; Oord et al., 2018; Caron et al., 2020; Grill et al., 2020). Representation learning, and specifically self-supervised learning, has also been used to achieve state-of-the-art generalization and sample efficiency results in RL on challenging control problems such as data efficient Atari (Schwarzer et al., 2020; 2021), DeepMind Control (Agarwal et al., 2021) and Procgen (Mazoure et al., 2020; Stooke et al., 2021; Raileanu and Fergus, 2021; Mazoure et al., 2021).Noteworthy instances of theoretically-motivated representation learning methods for RL include heuristic-guided learning  (Sun et al., 2018; Cheng et al., 2021), and random Fourier features (Nachum and Yang, 2021).

**Offline reinforcement learning**   When learning from a static dataset, agents should balance interpolation and extrapolation errors, while ensuring proper diversity of actions (i.e. prevent collapse to most frequent action in the data). Popular offline RL algorithms such as BCQ (Fujimoto et al., 2019), MBS (Liu et al., 2020c), and CQL (Kumar et al., 2020) rely on a behavior regularization loss (Wu et al., 2019) as a tool to control the extrapolation error. Some methods, such as F-BRC (Kostrikov et al., 2021) are defined only for continuous action spaces while others, such as MOReL (Kidambi et al., 2020) estimate a pessimistic transition model. The major issue with current offline RL algorithms such as CQL is that they are perhaps overly pessimistic for generalization purposes, i.e. CQL and MBS ensure that the policy improvement is well-supported by the batch of data. As we will show in our empirical comparisons, using overly conservative policy updates can prevent the representation from fully leveraging the information of the training dataset.

## 3   PROBLEM SETTING

### 3.1   PARTIALLY-OBSERVABLE MARKOV DECISION PROCESSES

A (infinite-horizon) partially-observable Markov decision process (POMDP, Murphy, 2000) $M$ is defined by the tuple $M = \langle \mathcal{S}, p_0, \mathcal{A}, p_{\mathcal{S}}, \mathcal{O}, p_{\mathcal{O}}, r, \gamma \rangle$, where $\mathcal{S}$ is a state space, $p_0 = \mathbb{P}[s_0]$ is the starting state distribution, $\mathcal{A}$ is an action space, $p_{\mathcal{S}} = \mathbb{P}[\cdot|s_t, a_t] : \mathcal{S} \times \mathcal{A} \to \Delta(\mathcal{S})$ is a transition function, $\mathcal{O}$ is an observation space, $p_{\mathcal{O}} = \mathbb{P}[\cdot|s_t] : \mathcal{S} \to \Delta(\mathcal{O})$[1] is an observation function, $r : \mathcal{S} \times \mathcal{A} \to [r_{\min}, r_{\max}]$ is a reward function and $\gamma \in [0, 1)$ is a discount factor. The system starts in one of the initial states $s_0 \sim p_0$ with observation $o_0 \sim p_{\mathcal{O}}(\cdot|s_0)$. At every timestep $t = 1, 2, 3, ..,$ the agent, parameterized by a policy $\pi : \mathcal{O} \to \Delta(\mathcal{A})$, samples an action $a_t \sim \pi(\cdot|o_t)$. The environment transitions into a next state $s_{t+1} \sim p_{\mathcal{S}}(\cdot|s_t, a_t)$ and emits a reward $r_t = r(s_t, a_t)$ along with a next observation $o_{t+1} \sim p_{\mathcal{O}}(\cdot|s_{t+1})$.

The goal of an RL agent is to maximize the cumulative rewards $\sum_{t=0}^{\infty} \gamma^t r_t$ obtained over the entire episode. Value-based off-policy RL algorithms achieve this by estimating the state-action value function under a target policy $\pi$:

$$Q^{\pi}(s_t, a_t) = \mathbb{E}_{\mathbb{P}_t^{\pi}}[\sum_{k=1}^{\infty} \gamma^k r(s_{t+k}, a_{t+k})|s_t, a_t], \ \forall s_t \in \mathcal{S}, a_t \in \mathcal{A} \tag{1}$$

where $\mathbb{P}_t^{\pi}$ denotes the joint distribution of $\{s_{t+k}, a_{t+k}\}_{k=1}^{\infty}$ obtained by executing $\pi$ in the environment.

---

[1]$\Delta(\mathcal{X})$ denotes the entire set of distributions over the space $\mathcal{X}$.

An important distinction from online RL is that, instead of sampling access to the environment, we assume access to a historical dataset $\mathcal{D}^\mu$ collected by logging experience of the policy, $\mu$, in the form $\{o_{i,t}, a_{i,t}, r_{i,t}\}_{i=1,t=1}^{i=N,t=T}$ where, for practical purposes, the episode is truncated at $T$ timesteps. Furthermore, we assume that the agent can only be trained on a limited collection of POMDPs $\mathcal{M}_{\text{train}} = \{M_i\}_{i=1}^m$, and its performance is evaluated on the set of test POMDPs $\mathcal{M}_{\text{test}}$. We assume that both $\mathcal{M}_{\text{train}}$ and $\mathcal{M}_{\text{test}}$ were sampled from a common task distribution and that every POMDP $M_i \in \mathcal{M} = \mathcal{M}_{\text{train}} \cup \mathcal{M}_{\text{test}}$ shares the same transition dynamics and reward function with $\mathcal{M}$ but has a different observation function $p_{i,\mathcal{O}}$. Importantly, since we perform control from pixels, we are in the POMDP setting (see Yarats et al., 2019) and therefore emphasize the difference between observations $o_t$ and corresponding states $s_t$ throughout the paper.

### 3.2 Representation learning

Previous works in the RL literature have studied the use of auxiliary signals to improve generalization performance. Among others, Liu et al. (2020a); Agarwal et al. (2021) define the similarity of two observations to depend on the distance between action sequences rolled out from that observation under their respective optimal policies. They achieve this by finding a latent space $\mathcal{Z} \subseteq \mathcal{S}$ in which the distance $d_{\mathcal{Z}}(z, z')$ for all $z, z' \in \mathcal{Z}$ is equivalent to distance between true latent states $d_{\mathcal{S}}(s, s')$ for all $s, s' \in \mathcal{S}$; the aforementionned works learn $\mathcal{Z}$ by optimizing action-based similarities between observations.

In practice, latent space $z$ is decoded from observation $o$ using a latent state decoder $f : \mathcal{O} \to \mathcal{Z}$ from observation $o_t$. Through the paper, we assume that all value functions have a linear form in the latent decoded state, i.e. $Q_\theta(o, a) = \theta_a^\top f_\psi(o) = \theta_a^\top z_\psi$, which agrees with our practical implementation of all algorithms. Within this model family, the ability of an RL agent to correctly decode latent states from unseen observations directly affects its policy, and therefore, its generalization capabilities. In the next section, we discuss why representation learning is important for offline RL, and how existing action-based similarity metrics fail to recover the true latent states for important families of POMDPs.

## 4 Motivating example

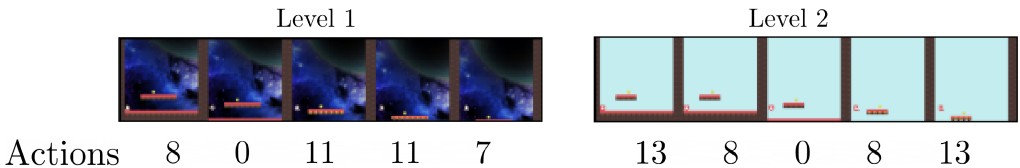

Figure 1: Two levels of the Climber game from the Procgen benchmark (Cobbe et al., 2020) with *near-identical* true latent states and *near-identical* value functions but *drastically different* action sequences.

Multiple recently proposed self-supervised objectives (Liu et al., 2020a; Agarwal et al., 2021) conjecture that observations $o_1 \in M_1, o_2 \in M_2$ that emit similar future action sequences under optimal policies $\pi_1^*, \pi_2^*$ should be decoded into nearby latent states $z_1, z_2$. While this heuristic can correctly group observations with respect to their true latent state in simple action spaces, it fails to identify similar pairs of trajectories in POMDPs with multiple optimal policies. For instance, two trajectories might visit an identical set of latent states, but have drastically different actions.

Fig. 1 shows one such example: two levels of the Climber game have a near-identical true latent state (see Appendix) and value function (average normalized mean-squared error of 0.0398 across episode), while having very different action sequences from a same PPO policy (average total variation distance of 0.4423 across episode). The problem is especially acute in Procgen, since the PPO policy is high-entropy for some environments (see Fig. 4), i.e. various levels can have multiple drastically different near-optimal policies, and hence fail to properly capture observation similarities.

In this scenario, assigning observations to a similar latent state by value function similarity would yield a better state representation than reasoning about action similarities. In a POMDP with a

different structure, grouping representations by action sequences can be optimal. So how do we unify these similarity metrics under a single framework?

In the next section, we use this insight to design a general way of improve representation learning through self-supervised learning of discounted future behavior.

## 5 METHOD

We propose to measure a generalized notion of future behavior similarity using generalized value functions, as defined by the corresponding cumulant function. The choice of cumulant determines which components of the future trajectory are most relevant for generalization.

### 5.1 QUANTIFYING FUTURE BEHAVIOR WITH GVFs

An RL agent's future discounted behavior can be quantified not only by its the value function, but other auxiliary signals, for example, by its observation occupancy measure, known as successor features (Dayan, 1993; Barreto et al., 2016). The choice of the examined signal quantifies the properties the agent will exhibit in the future, such as accumulated returns, or observation visitation density. See Thm. 2 in the Appendix for the connection between successor features and interpolation error in our method.

Following the work of Sutton et al. (2011), we can broaden the class of value functions to any kind of cumulative discounted signal, as defined by a bounded cumulant function $c : \mathcal{O} \times \mathcal{A} \to \mathbb{R}^d$, s.t. $|c(o, a)| \leq c_{\max}$ for $c_{\max} = \sup_{o,a \in \mathcal{O} \times \mathcal{A}} c(o, a)$. While typically cumulants are scalar-valued functions (e.g. reward), we also make use of the vector-valued case for learning the successor features (Barreto et al., 2016), in which case the norm of $c(o, a)$ is bounded.

**Definition 1 (Generalized value function)** *Let $c$ be any bounded function over $\mathbb{R}^d$, let $\gamma \in [0, 1]$ and $\mu$ any policy. The generalized value function is defined as*

$$G^\mu(o_t) = \mathbb{E}_{\mathbb{P}_t^\mu}[\sum_{k=1}^{\infty} \gamma^k c(o_{t+k}, a_{t+k})|o_t] \tag{2}$$

*for any timestep $t \geq 1$ and $o_t \in \mathcal{O}$.*

Since, in our case, we can learn $G^\mu$ for each distinct POMDP $M_i$ for the dataset $\mathcal{D}^\mu$, we index the GVF using the POMDP index, i.e. $G_i^\mu = \texttt{LearnGVF}(c, \mathcal{D}^\mu, i)$ (in practice, the learning is parallelized).

---

**Algorithm 1:** $\texttt{LearnGVF}(c, \mathcal{D}^\mu, i, \theta^{(0)}, J, \alpha, \gamma)$: Offline estimation of GVF $\hat{G}_i^\mu$

**Input** : Cumulant function $c$, dataset $\mathcal{D}^\mu$, POMDP label $i$, initial parameters $\theta^{(0)}$, target parameters $\tilde{\theta}$, latent state decoder $f$, iterations $J$, learning rate $\alpha$, discount $\gamma$

1 **for** $j = 1, .., J$ **do**
2    $o, a, o' \sim \mathcal{D}[i]$; // Sample transition from POMDP $i$
3    $c \leftarrow c(o, a)$;
4    $o \leftarrow \text{random crop}(o)$;
5    $z, z' \leftarrow f(o), f(o')$;
6    $\theta^{(j)} \leftarrow \theta^{(j-1)} - \alpha \nabla_{\theta^{(j-1)}} (G_{\theta^{(j-1)}}(z) - c - \gamma G_{\tilde{\theta}^{(j-1)}}(z'))^2$ ;
7    Update target parameters $\tilde{\theta}$ with $\beta$ of online parameters $\theta$;

---

### 5.2 MEASURING DISTANCES BETWEEN GVFs OF DIFFERENT POMDPs

Examining the difference between future behaviors of two observations quantifies the exact amount of expected behavior change between these two observations. Using the GVF framework, we could compute the distance between $o_1 \in M_1$ and $o_2 \in M_2$ by first estimating the latent state with $z = f(o)$ using a (learned) latent state decoder $f$, and then evaluating the distance

$$d_\mu(o_1{}^i, o_2{}^j) = |G_i^\mu(f(o_1)) - G_j^\mu(f(o_2))| \quad i, j = 1, 2, .., \tag{3}$$

a measure of dissimilarity that can then be used in a contrastive loss.

However, the distance between GVFs from two different POMDPs can have drastically different scales: $|G_1^\mu(o_1) - G_2^\mu(o_2)| \leq \frac{c_{1,\max}^\mu + c_{2,\max}^\mu}{1-\gamma}$, making point-wise comparison meaningless. The issue is less acute for cumulants which induce a unnormalized density estimate (e.g. indicator functions for successor representation), and more problematic when the cumulant incorporates the extrinsic reward function. To avoid this problem, we suggest performing a comparison based on order statistics.

A robust distance estimate between GVF signals across POMDPs can be obtained by looking at the cumulative distribution function of $G_i$ denoted $F_i(g) = \mathbb{P}[G_i(o_t) \leq g]$ for all $o_t \in \mathcal{O}$. $G_i$ is a deterministic GVF with the set of discontinuity points of measure 0, and as such $F_i$ can be understood through the induced state distribution $\mathbb{P}_t^\mu$ (using continuous mapping theorem from Mann and Wald (1943)). It can be estimated from $n$ independent and identically distributed samples of $\mathcal{D}^\mu$ as

$$\hat{F}_i(g) = \frac{1}{n}\sum_{i=1}^n \mathbb{1}_{G_i < g}, G_i = \texttt{LearnGVF}(c, \mathcal{D}^\mu, i), \ g \in \left[ -\frac{c_{i,\max}}{1-\gamma}, \frac{c_{i,\max}}{1-\gamma} \right] \tag{4}$$

and its inverse, the empirical quantile function (van der Vaart, 1998)

$$\hat{F}_i^{-1}(p) = \inf\{g \in \left[ -\frac{c_{i,\max}^\mu}{1-\gamma}, \frac{c_{i,\max}^\mu}{1-\gamma} \right] : p \leq F_i(g)\}, \ p \in [0,1] \tag{5}$$

We use the empirical quantile function to partition the range of all GVFs into $K$ quantile bins, i.e. disjoint sets with identical size where the set corresponding to quantile $k$ is defined as $I_i(k) = \{o \in M_i : F_i^{-1}(\frac{k}{K}) \leq G_i^\mu(o) \leq F_i^{-1}(\frac{k+1}{K})\}$ and it's aggregated version as $I(k) = \cup_{i=1}^m I_i(k)$.

Importantly, we augment the dataset $\mathcal{D}^\mu$ with observation-specific labels, which correspond to the index of the quantile bin into which the GVF $G$ of an observation $o \in M_i$ falls into:

$$l_i(o) = \max_k \mathbb{1}_{o \in I_i(k)} \tag{6}$$

These self-supervised labels are then used in a multiclass InfoNCE loss (Oord et al., 2018), which is a variation of metric learning with respect to the quantile distance defined above (Khosla et al., 2020; Song and Ermon, 2020).

## 5.3 SELF-SUPERVISED LEARNING OF GSFS

After augmenting the offline dataset with observation labels, we use a simple self-supervised learning procedure to minimize distance in the latent representation space between observations with identical labels.

First, the observation $o$ is encoded using a non-linear encoder $f_\psi : \mathcal{O} \to \mathcal{Z}$ with parameters $\psi$ into a latent state representation $z = f_\psi(o)$[2]. The representation $z$ is then passed into two separate trunks: 1) a linear matrix $\theta_a$ which recovers the state-action value function $Q_\theta(o, a) = \theta_a^\top z$, and 2) a non-linear projection network $h_\theta : \mathcal{Z} \to \mathcal{Z}$ with parameters $\theta_h$ to obtain a new embedding, used for contrastive learning.

The projection $h(z)$ is then used used in a multiclass InfoNCE loss (Oord et al., 2018; Song and Ermon, 2020) where a linear classifier $\mathbf{W} \in \mathbb{R}^{|\mathcal{Z}| \times K}$ aims to correctly predict the observation labels (i.e. quantile bins $k = 1, 2, .., K$) from $h(z)$:

$$\ell_{\text{NCE}}(\theta_h, \psi, \mathbf{W}) = -\mathbb{E}_{o \sim \mathcal{D}^\mu}\left[ \sum_{k=1}^K \mathbb{1}_{l(o)=k}\text{LogSoftmax}[\mathbf{W}^\top h(f_\psi(o))/\tau]_k \right], \tag{7}$$

where $\tau > 0$ is a temperature parameter.

Our empirical findings suggest that this version of the loss is more stable than other multi-class contrastive losses (see Appendix 7.3).

---

[2]This encoder is different from the one used to evaluate the GVFs.

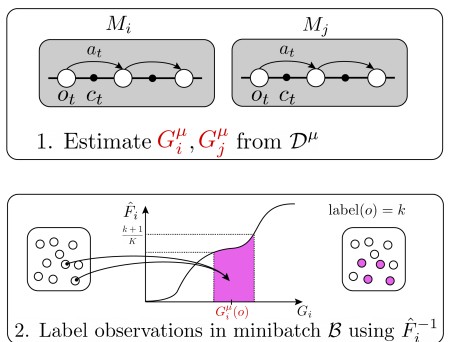

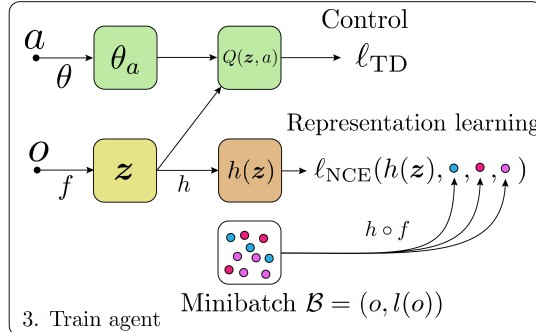

Figure 2: Schematic view of GSF : the offline dataset $\mathcal{D}^\mu$ is used to estimate POMDP-specific GVFs wrt some cumulant function $c$, whose quantiles are then used to label each observation in the dataset. These labels are then used in a multi-class contrastive learning procedure along with offline RL learning.

## 5.4 ALGORITHM

Our method relies on the approximation oracle `LearnGVF`, to produce GVF estimates, later used in the contrastive learning phase.

Since we are concerned with the offline RL setting, we add our auxiliary loss on top of Conservative Q-learning (CQL, Kumar et al., 2020), a strong baseline. CQL is trained using a linear combination of Q-learning (Watkins and Dayan, 1992; Ernst et al., 2005) and behavior-regularization:

$$\ell_{\mathrm{CQL}}(\theta) = \mathbb{E}_{o,a,r,o'\sim\mathcal{D}^\mu}[(r+\gamma\max_{a'\in\mathcal{A}}Q_{\tilde{\theta}}(o',a')-Q_\theta(o,a))^2]+\lambda\mathbb{E}_{s\sim\mathcal{D}^\mu}[LSE(Q_\theta(o,a))-\mathbb{E}_{a\sim\mu}[Q_\theta(o,a)]], \tag{8}$$

for $\lambda \geq 0$, $\tilde{\theta}$ target network parameters[3] and $LSE$ being the log-sum-exp operator [4].

---

**Algorithm 2:** GSF : Offline RL with future behavior observation matching

**Input** : Dataset $\mathcal{D} \sim \mu$, initialized Q-function $Q_\theta$ with encoder $f_{\theta_f}$ and action weights $\theta_a$, per-POMDP set of GVFs $\mathcal{G} = \{G_i^\mu\}_{i\in}$, state projection network $h_\psi$, epoch number $J$, number of POMDPs $m$, number of quantiles $K$, temperature parameter $\tau$, exponential moving average parameter $\beta$

1 **for** *epoch* $j = 1, 2, .., J$ **do**
2   **for** *minibatch* $\mathcal{B} \sim \mathcal{D}$ **do**
    `/* Data augmentation on observation                    */`
3     $o \leftarrow$ random crop$(o)$ for all $o \in \mathcal{B}$;
4     $z \leftarrow f_\psi(o)$ for all $o \in \mathcal{B}$ ;
    `/* Update CQL agent                                    */`
5     Update $\theta_a, \psi$ using $\nabla_{\theta_a,\psi}\ell_{\mathrm{CQL}}(\theta)$ ;
    `/* Compute $\mathcal{G}$ quantiles                       */`
6     **for** *POMDP* $M_i = 1, 2, .., m$ **do**
7       Estimate $\hat{F}_i^{-1}$ of $G_i^\mu$ from $\mathcal{B}$ ;
8       **for** *observation* $o \in \mathcal{B} \cap M_i$ **do**
9         $l(o) \leftarrow k$ if $\hat{F}_i^{-1}(\frac{k}{K}) \leq G_i^\mu(o) \leq \hat{F}_i^{-1}(\frac{k+1}{K})$ ;
    `/* Update encoder and projection network               */`
10     Update $\theta_h, \psi, \mathbf{W}$ using $\nabla_{\theta_h,\psi,\mathbf{W}}\ell_{\mathrm{NCE}}(\theta_h, \psi, \mathbf{W})$ computed with $z, l(o)$ and $\tau$ ;
11     Update CQL agent's target network with $\beta$ of online parameters $\psi, \theta$;

---

Alg. 2 summarizes the learning procedure for GSF as implemented on top of a CQL agent for a discrete action space. In our experiments, all baselines use random crops as data augmentation.

---

[3]A copy of $\theta$ updated solely using an exponential moving average (see Appendix).
[4]https://en.wikipedia.org/wiki/LogSumExp

**Connection to existing methods**   Our framework is able to recover objectives similar to those of prior works by carefully designing the cumulant function.

- **Cross-State Self-Constraint (CSSC, Liu et al., 2020a)**: In CSSC, observations $o_1, o_2$ are considered similar if they have identical future action sequences of length $K$ under some fixed policy; a total of $|\mathcal{A}|^K$ distinct classes are possible. This approach can be approximated in our framework by picking $c(o_t, a_t) = \mathbb{1}_{a_t}(a), \ \forall a \in \mathcal{A}$. The problem reduces to a $|\mathcal{A}|^{T-t}$-way classification problem for observations of timestep $t$, which GSF approximates using $K$ quantiles.

- **Policy similarity embedding (PSE, Agarwal et al., 2021)**: PSEs balance the distance between local optimal behaviors and long-term dependencies in the transitions, notably using $d_{\mathrm{TV}}$. If we consider the space of Boltzmann policies $\pi_{\mathrm{Boltzmann}}$ with respect to an POMDP-specific value function $Q$, then choosing $c(o_t, a_t) = r(s_t, a_t)$ in GSF will effectively compute the distance between unnormalized policies.

## 5.5   CHOICE OF NUMBER OF QUANTILES $K$

How should the number of quantiles $K$ be set, and what is the effect of smaller/ larger values of $K$ on the observation distance? Thm. 1 highlights a trade-off when choosing the number of quantiles bins empirically.

**Theorem 1** *Let $G_1$, $G_2$ be generalized value functions with cumulants $c_1, c_2$ from respective POMDPs $M_1, M_2$, $K$ be the number of quantile bins, $n_1, n_2$ the number of sample transitions from each POMDP. Suppose that $\mathbb{P}[\sup_{t=1,2,...} |c_1(o_{1,t}, \mu(o_{1,t})) - c_2(o_{2,t}, \mu(o_{2,t}))| > {(1-\gamma)\varepsilon}/{\gamma}] \leq \delta$. Then, for any $k = 1, 2, .., K$ and $\varepsilon > 0$ the following holds without loss of generality:*

$$\mathbb{P}\left[\sup_{o_1, o_2 \in I(k)} |G_1(o_1) - G_2(o_2)| > 3\varepsilon\right] \leq 2e^{-2n_1\varepsilon^2/4} + p(n_1, K, \varepsilon) + \delta \tag{9}$$

*where*

$$p(n, K, \varepsilon) = \mathbb{P}\left[\sup_{k=1,2,..,K} \left|\hat{F}_1^{-1}\left({k+1}/{K}\right) - \hat{F}_1^{-1}\left({k}/{K}\right)\right| > \varepsilon\right] \tag{10}$$

.
The proof can be found in the Appendix Sec. 7.2. For POMDP $M_1$, the error decreases monotonically with increasing bin number $K$ (second term) but the variance of bin labels depends on the number of sample transitions $n_1$ (first term). The inter-POMDP error (third term) does not affect the bin assignment. Hence, choosing a large $K$ will amount to pairing states by rankings, but results in high variance, as orderings are estimated from data and each bin will have $n = 1$. Setting $K$ too small will group together unrelated observations, inducing high bias.

## 6   EXPERIMENTS

Unlike for single task offline RL (Fu et al., 2020), most works on zero-shot generalization from offline data either come up with an *ad hoc* solution suiting their needs, e.g. (Ball et al., 2021), or assess performance on benchmarks that do not evelute generalization across observation functions (e.g., Yu et al., 2020). To accelerate progress in this field, we devised the offline Procgen benchmark, an offline RL dataset to directly test for generalization of offline RL agents across observation functions[5].

**Offline Procgen benchmark**   We evaluate the proposed approach on an offline version of the Procgen benchmark (Cobbe et al., 2020), which is widely used to evaluate zero-shot generalization across complex visual perturbations. Given a random seed, Procgen allows to sample procedurally generated level configurations for 16 games under various complexity modes: "easy", "hard" and "exploration". The dataset is obtained as follows: we first pre-train a PPO (Schulman et al., 2017) agent for 25M timesteps on 200 levels of "easy" distribution for each environment[6] ("easy" mode is widely used to test generalization capabilities (Cobbe et al., 2020; Raileanu and Fergus, 2021; Mazoure et al., 2021)). All agents use the IMPALA encoder architecture (Espeholt et al., 2018), which has enough parameters to allow better generalization performance, compared to other models (e.g., Mnih et al., 2015).

---

[5]The benchmark will be open-sourced.

[6]We use the TFAgents' implementation (Guadarrama et al., 2018)

**Results** We compare the zero-shot performance on the entire distribution of "easy" POMDPs for GSF against that of strong RL and representation learning baselines: behavioral cloning (BC) - to assess the quality of the PPO policy, CQL (Kumar et al., 2020) - the current state-of-the-art on multiple offline benchmarks which balances RL and BC objectives, CURL (Srinivas et al., 2020), CTRL (Mazoure et al., 2021), DeepMDP (Gelada et al., 2019) - which learns a metric closely related to bisimulation across the MDP, Value Prediction Network (VPN, Oh et al., 2017) - which combines model-free and model-based learning of values, observations, next observations, rewards and discounts, Cross-State Self-Constraint (CSSC, Liu et al., 2020a) - which boosts similarity of observations with identical action sequences, as well as Policy Similarity Embeddings (Agarwal et al., 2020), which groups observation representations based on distance in optimal policy space.

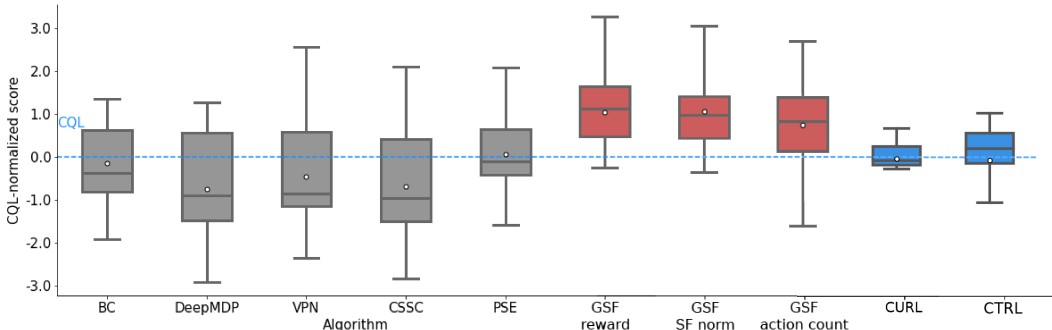

Figure 3: Returns on the offline Procgen benchmark (Cobbe et al., 2020) after 1M training steps. Boxplots are constructed over 5 random seeds and all 16 games; each method is normalized by the per-game median CQL performance. White dots represent average of distribution.

Fig. 3 shows the performance of all methods over 5 random seeds and all 16 games on the offline Procgen benchmark after 1 million training steps. Per-game average scores for all methods can be found in Tab. 2 (Appendix). The scores are standardized per-game using the downstream task's (offline RL) performance, in this case implemented by CQL. It can be seen that GSF performs better than other offline RL and representation learning baselines.

Using different cumulants functions can lead to different label assignments and hence different similarity groups. Fig. 3 examines the performance of GSF with respect to 3 cumulants: 1) $r(s_t, a_t)$, rewards s.t. GSF learns the policy's $Q^\mu$-value, 2) $\mathbb{1}_{o_t}(o)$, the successor representation[7] (Dayan, 1993; Barreto et al., 2016) s.t. GSF learns induced distribution over $\mathcal{D}^\mu$ (Machado et al., 2020) and 3) $\mathbb{1}_{a_t}(a)$, action counts, s.t. GSF learns discounted policy. While rewards and successor feature cumulant choices leads to similar performance, using action-based distance leads to larger variance.

## 7 DISCUSSION

In this work we proposed GSF, a novel algorithm which combines reinforcement learning with representation learning to improve zero-shot generalization performance on challenging, pixel-based control tasks. GSF relies on computing the similarity between observation pairs with respect to any instantaneous accumulated signal, which leads to improved empirical performance on the newly introduced offline Procgen benchmark. Theoretical results suggest that GSF 's hyperparameter choice depends on a trade-off between finite sample approximation and extrapolation error.

While our work answered some questions regarding zero-shot generalization in offline RL, some questions persist: can GVF-based distances be included in a contrastive objective without the need for quantile discretization (perhaps through re-scaling or order statistics)? Can the cumulant function be chosen *a priori* for a specific task structure other than Procgen, and shown to lead to optimal representations?

---

[7]In the continuous observation space, we learn a $d$-dimensional successor feature vector $z_\psi$ via TD and computing the quantiles over $||z_\psi||_1$.

## REPRODUCIBILITY STATEMENT

To ensure reproducibility of our work, we attach the source code to the submission, provide all proofs of both theorems in the appendix (with assumptions) and will open-source the offline Procgen benchmark.

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
