# OpenReview forum: "Improving zero-shot generalization in offline reinforcement learning using generalized similarity functions"
_ICLR.cc/2022/Conference — ICLR 2022 Submitted_

### Official Review · Reviewer_f74c · 2021-11-02

**Correctness:** 3
**Technical Novelty And Significance:** 3
**Empirical Novelty And Significance:** 2
**Recommendation:** 5
**Confidence:** 2

**Main Review:**

Strength:

- RL generalization in an offline setting is an important and interesting topic for our community.
- The approach is well-motivated by the example in Section 4.
- The proposed GVF is a general framework that can recover objectives of existing works.
- The proposed approach outperforms baselines such as CQL, CCSC and PSE.

Weakness:

- The reviewer found the approach section is not easy to follow. Some technical details seem missing. Please see detailed comments below:
    - The latent state decoder f seems to be a critical part of the proposed pipeline. Could you elaborate on how you learn this decoder?
    - Definition 1 states that the cumulant function c can be any bounded function over R^d. It is unclear to the reviewer how to select the cumulant function c. What cumulant function is used in the experiments. Does each task require a different cumulant function?


- In addition, the reviewer found the notations confusing. Please see detailed comments below:
   - In line 2 of Algorithm 1, what does D[i] represent? What is the difference between D[i] and D^{\mu}
   - In line 6 of algorithm 1, does G represent GVF? If yes, why does it take a latent representation as input? According to Definition 1, shouldn’t GVF take an observation o as input?
   - "Target parameter" and "online parameter" are not formally defined
   - $\beta$ in line 7 of algorithm 1 is undefined?


- Experiments
  - The authors discussed F-BRC (Kostrikov et al., 2021) and MOReL (Kidambi et al., 2020), which are state-of-the-art offline RL approaches, in Section 2. However, there is no comparison with the aforementioned existing works provided. The experimental section could be more convincing if the comparison with F-BRC and MOReL is provided.

  - Recent works [a] show that data augmentation only could significantly improve the generalization capability of RL agents. The proposed approach also uses data augmentation during training (Line 4 of Algorithm 1).  It would be interesting to see an ablation study on the data augmentation and the proposed GVF.

[a] Reinforcement Learning with Augmented Data, Laskin et al. NeurIPS 2020


**Summary Of The Paper:**

This paper studied  zero-shot generalization in an offline reinforcement learning setting. The authors proposed to improve the generalization via a better representation learning. Specifically, the authors hypothesized that observation with similar future behaviors should be assigned to similar representations. To this end, the generalized similarity function (GSF) that aggregates latent representations with the future behavior is proposed. The proposed approach is evaluated on an offline version of Procgen. The proposed approach outperforms baselines across different tasks in Procgen.


**Summary Of The Review:**

The reviewer thinks the topic of this paper is significant. The approach seems interesting. However, the reviewer has some concerns about the experimental results and the clarity of the presentation.

---

> ### Author Response · Authors · 2021-11-16
> **Response**
>
> We thank you for the constructive comments. Below are the answers to your concerns:
>
> 1. **Regarding latent state decoder:** We apologize for the abuse of terminology here and will clarify it in future revisions. The latent state decoder, is a convolutional neural network (at least for the offline Procgen benchmark), which takes in 64 x 64 x 3 pixel observations and maps them onto a latent vector of size 256, called *representation*. The mapping operation can be thought of as “decoding the latent state $z$ from observation o”, hence the dual nomenclature. In the case of GSFs, the encoder is trained using the cross-entropy loss jointly with the CQL loss from Eq.8; for baselines, it is trained using their respective auxiliary losses + CQL loss.
>
> 2. **Regarding cumulant functions:** While, theoretically, the cumulant function can be any real-valued signal satisfying the definition in the second paragraph of Sec.5.1, in practice, we studied 3 cumulants: (i) rewards, which allows GSF to recover the value function, (ii) $\ell_1$ norm of the successor feature vector trained with TD, which allows GSF to recover pseudocounts (see "Count-Based Exploration with the Successor Representation" by Machado et al. 2020) and (iii) action counts for a specific state, allowing GSF to recover the extension of the CSSE objective to infinite-length n-grams. The best performing cumulant on the offline Procgen benchmark was the reward, due to Procgen exhibiting the property highlighted in Fig.1. However, for a different benchmark, a different choice of cumulant function might work better.
>
> 3. **Regarding F-BRC and MOReL:** The F-BRC method is a good alternative to CQL (although it performs roughly similarly to CQL on D4RL tasks), but is not applicable to discrete action spaces, of which Procgen is one (the derivative of the categorical policy is undefined with respect to the actions, and hence the Fisher divergence is undefined). The issue with MOReL is that no code is (yet) publicly available. Moreover, model-based methods are not typically used on the Procgen zero-shot generalization benchmark (see Cobbe et al., 2020), due to the difficulty of learning a single model per level for hundreds of levels.
>
> 4. **Regarding data augmentation:** While GSF uses data augmentation (specifically, random crops), so do all the other baselines for fair comparison (see last sentence of page 7 of main paper).
>
> ###### Minor issues (all addressed in revision):
>
> 1. You’re right, the notation $\mathcal{D}[i]$ was not formally introduced, but it means “data from $\mathcal{D}$ belonging to POMDP $i$”, which is a similar property that PSEs deal with.
>
> 2. This is an abuse of notation: the GVF $G$ is a mapping from $\mathcal{O}$, meaning that $G(o_t)$ is valid. However, the way that $o_t$ is used under-the-hood is identical to a classical value function: first map $o_t$ onto a representation $z_t=f(o_t)$ using an encoder/latent state decoder, then pass $z_t$ through an MLP to obtain $G(o_t)$.
>
> 3. We added a proper definition in the revision (page 7 + Appendix).

---

### Official Review · Reviewer_nnNd · 2021-11-02

**Correctness:** 3
**Technical Novelty And Significance:** 3
**Empirical Novelty And Significance:** 2
**Recommendation:** 6
**Confidence:** 4

**Main Review:**

Strenght:
* The paper is clearly written and the various choices are reasonably justified
* The proposition of an offline version of Procgen is a valuable asset for the offline RL research community
* the approach of quantization in this context is novel to the best of my knowledge
* The experimental results seem significant for the considered benchmark dataset
* the introduction of generalization value function seem novel and pertinent

Weakness:
* The comparison to cURL (https://arxiv.org/abs/2004.04136) is missing, especially considering the fact that this approach is also leveraging metric learning in latent state space.
* The D4RL dataset would have been useful to compare too, especially to evaluate over the latent recent corpus of approaches of offline RL.
* The approach relies on PPO, which is an online RL algorithm, and CQL which is a conservative approach of Q-Learning. Maybe it would have been interesting to evaluate how the method behaves with SAC also.

**Summary Of The Paper:**

The paper addresses the problem of generalization and representation learning in reinforcement learning.
Concretely, the authors propose an approach of self-supervision for improving offline-RL based on similarity learning over the considered state space.
In this context, the concept of generalized value function is introduced.
The similarity measure is defined as a classification problem over quantized latent space.
The paper is evaluated in an offline setting of Procgen which is a dataset for multi-task reinforcement learning generalization evaluation.
More particularly, the proposed approach claims to be particularly efficient in improving zero-shot generalization performance on one offline RL benchmark, offline Procgen.
The offline Procgen dataset is a second contribution of the paper.

**Summary Of The Review:**

The paper is addressing an important problem of sequential decision learning research.
The proposing is interesting while novelty to existing approach could be improved.
The proposition of a new dataset for zero-shot generalization of sequential decision-making in an offline context is valuable.
The comparison on another popular benchmark like D4RL and against the cURL approach would have been appreciated.
The introduction of generalized value function seems novel and pertinent in the context of self-supervised reinforcement learning and is clearly illustrated in the proposal of this work and could be valuable to the state of the art.

---

> ### Author Response · Authors · 2021-11-16
> **Response**
>
> Thank you for the insightful comments and suggestions; our response is below:
>
> 1. **Regarding cURL:** Indeed, cURL is among the most widely known contrastive state representation learning objectives in RL. We ran cURL over 5 random seeds, and our GSF algorithm outperforms cURL according to these results (see updated Fig.3 in revision). Note that, however, cURL doesn’t use any temporal information (which GSF, PSEs and CSSC do use), and hence its performance is potentially suboptimal as a result.
>
> 2. **Regarding D4RL:** The D4RL dataset is of course a widely popular benchmark, but unfortunately it is not directly applicable in our setting, as it tests for single-task sample efficiency instead of zero-shot generalization to new task variants.
>
> 3. **Regarding PPO vs SAC comparison:** Yes, our approach relies on PPO to *collect* the offline dataset $\mathcal{D}^\mu$, but that is the only time we ever use an online RL algorithm, similarly to other offline RL benchmarks (e.g. “D4rl: Datasets for deep data-driven reinforcement learning” by Fu et al. 2020). We are not sure whether you are suggesting to collect data from Procgen using SAC instead of PPO, or compare against an offline SAC baseline? For the former - PPO is a well-established baseline on Procgen which is used by the overwhelming majority of algorithms developed for solving Procgen (see “Leveraging Procedural Generation to Benchmark Reinforcement Learning” Fig.6 by Cobbe et al. 2020). For the latter - offline SAC is essentially CQL with the regularization term set to 0 and with an entropy boosting term. In our experiments, we have tried various values of the CQL regularization weight but the best performing one (for zero-shot generalization specifically) was $\lambda=1$ (see hyperparameter table).

---

### Official Review · Reviewer_vyyv · 2021-11-03

**Correctness:** 4
**Technical Novelty And Significance:** 3
**Empirical Novelty And Significance:** 3
**Recommendation:** 6
**Confidence:** 4

**Main Review:**

The proposed method is an interesting method for defining 'similarity' which can then be used for contrastive learning. The paper is well written and does not appear to have any technical or theoretical flaws. The topic of generalization in RL is relevant and the proposed method is a valuable contribution to this area.

One disadvantage of this method is that it requires the ability to pre-traing the generalized value function, hence the authors' decision to only apply it to offline RL. This leads to the, imo, main weakness of the paper: While the authors evaluate the method against a range of relevant baselines, they only do so in one domain, namely their newly proposed offline procgen benchmark. Hence the results will depend heavily on the authors effort to not only tune the hyperparameters of their own method, but also those of the baselines. Unfortunately, there is no information in the paper how the hyperparameters were chosen (they do provide the final hyperparameters of their own method only), so it's hard to evaluate how much one should trust the experimental evidence.

Hence, I think the paper could be strengthend significantly by evaluating on additional domains and/or providing detailed information about they applied hyperparameter search.

Question/Nitpick:
* Can you still call it contrastive loss? Eq. (7) just seems to be classification?

**Summary Of The Paper:**

The authors build on previous work which regularized states with similar value-function or future action sequences to have similar representations, by instead looking at successor features, i.e. the accumulated value of future cumulants. To deal with continuous value function whose scale can vary accross environments, they apply a quantile-binning technique. They also propose an offline version of Procgen as benchmark.

**Summary Of The Review:**

Overall interesting method, but more experimental detail needed.

---

> ### Author Response · Authors · 2021-11-16
> **Response**
>
> Thank you for the remarks. We have tested our proposed method on a custom offline RL benchmark due to the lack of generalization benchmarks in the existing offline RL literature. The offline Procgen benchmark aims specifically to test the generalization capabilities of RL agents without any further adaptation, by providing i) a similar structure between train and test sets which can be leveraged by the learner without adaptation at test time and ii) sufficient overlap of the training set with the optimal solution via the PPO policy at 25M frames corrupted with $\varepsilon$-greedy noise. We believe that this benchmark specifically helps us test the capabilities which allow GSF (and other baselines) to generalize well to unseen game levels. Indeed, if you are aware of an existing offline RL benchmark that aims to test zero-shot perceptual generalization, we can try to incorporate those experiments if time permits.
>
> 1. **Regarding hyperparameter tuning:** We added these details to the appendix. GSF’s parameters for the CQL regularization were taken to be identical to all other baselines (to ensure fairness and reasonable computational complexity). The other tuned hyperparameters were the baseline-specific auxiliary loss coefficients (e.g. GSF in our case, 1-Wasserstein for PSEs, etc) - the rest were taken to be default values (GSF parameters for temperature were taken to be default ones from self-supervised literature such as MoCo). Most of the baselines (e.g. CSSC and PSEs) were tested in an online setting near-identical to the offline Procgen benchmark, and hence the structural design choices should be expected to transfer reasonably well.
> 2. **Regarding nitpicks and terminology of “contrastive” learning:** We agree that the term is overly broad, especially that the final loss, which is used to learn GSFs, is a cross-entropy term. However, notice that multi-class InfoNCE, a contrastive objective, can be re-written (in one of its forms) as a categorical cross-entropy function (see “Momentum Contrast for Unsupervised Visual Representation Learning” by He et al. 2020) and so it is commonly referred to as contrastive in recent literature. On top of that, we studied two versions of the contrastive objective: one using categorical cross-entropy, as well as an aggregated InfoNCE loss. However, if the consensus among reviewers is that the naming is misleading, we will remove the “contrastive” characterization of GSF through the paper. We have already replaced some instances of "contrastive learning" with "self-supervised learning" in the revision.

---

> > ### Comment · Reviewer_vyyv · 2021-11-28
> > **Thank you for your response**
> >
> > Thank you for your response, I have raised my score.
> >
> > **Adding the information about hyperparameters of baselines:**. Thank you! As computation allows, I would encourage a more thorough hyperparameter search, especially if an additional environment is infeasible due to lack of suitable ones. Due to lack of comparability with previous work, the stronger the case for strong baselines can be made, the better.
> >
> > **Contrastive learning:** I don't have a strong opinion on the naming (hence "nitpick/question"). In the literature that I am familiar with, the "contrastive" nature of the InfoNCE loss is that the cross-entropy term was over "picking among inputs", not directly predicting known labels.

---

### Official Review · Reviewer_8hYV · 2021-11-07

**Correctness:** 2
**Technical Novelty And Significance:** 2
**Empirical Novelty And Significance:** 2
**Recommendation:** 5
**Confidence:** 3

**Main Review:**

The idea of aggregating observations based on behavior similarity has already been explored in some papers, but the approach proposed in this paper seem to be new.
While the paper discusses its relation to CSSC and PSE and compares with these two methods in the experiments too, it misses a discussion or empirical comparison with CTRL. In addition, it is not clear whether the proposed algorithm's advantage over CSSC and PSE in the experiments is due to the use of a better aggregation method, which is the main claimed novelty. Can other aggregation methods be combined with CQL?

The problem setup and the motivation are not clear and often confusing.
* There is a consistent confusion of MDP and POMDP throughout the paper. For example, in the introduction, the paper claims improving zero-shot generalization across a family of POMDPs, but the cited reference is Puterman's article on MDP. It then sometimes called the models POMDPs or MDPs.
* In the historical dataset ${\cal D}^{\mu}$, what is required for $\mu$?
* What is a concrete example of the problem setting described in Section 3.1? In the motivating example, it seems the first images for Level 1 and Level 2 should be considered similar, but it is not clear why. The main paper states that they have near identical true latent states and value functions, but didn't explain why.
* It's not clear what the different MDP/POMDP models are in the experiments.

The technical writing for the proposed approach requires a careful revision too. Here are some issues:
- The term cumulant function is confusing as cumulant function is a well-known concept in statistics.
- Section 3.1: for POMDP, a policy is usually maps a belief to an action, not an observation to an action. Also note that the Q-function in Eq. (1) is a function of the state and the action, and thus not directly computable because the state is not observed.
- $c$'s output may be a vector, and $|c(o, a)|$ and $\sup_{o, a} c(o, a)$ need to be defined.
- Alg. 1: D[i] in line 1 is undefined; line 6 works only for scalar c; line 7 is not clear.
- Eq. (3): RHS depend on $i$ and $j$, but LHS does not. Where is the $d_{\mu}$ used?
- $c_{1,\max}^{\mu}$ and $c^{\mu}_{2, max}$ are undefined.
- In $F_{i}(g) = \mathbb{P}[G_{i}(o_{t}) \le g]$, is $o_{t}$ treated as a random variable? What is the underlying distribution used? Is $G_{i}$ the same as $G_{i}^{\mu}$?
- Eq. (4): what does $G_{i} \sim {\cal D}^{\mu}$ mean? Note that ${\cal D}^{\mu}$ is defined as a set of $(o, a, r)$ triplets.
- Eq. (6): $i$ appears on the RHS but not on the LHS.
- Eq. (7): $l(z)$ and $o \sim {\cal D}$ undefined. Does ${\cal D}$ refer to ${\cal D}^{\mu}$?
- Eq. (8): $(s, a, r) \sim {\cal D}^{\mu}$?
- Alg. 1: should line 10 be inside the for loop? Line 11 is not clear.

The representation learned by DeepMDP is closely connected to bisimulation, but it doesn't learn bisimulation relations.

**Post-rebutall**

I appreciate the clarification and the additional results. I've raised my score in view of these changes. It'd be helpful to further stengthen the experiments and improve the writing. The writing is generally clear now, but there are still quite a few issues (e.g., $o_{1}$ and $o_{2}$ in Eq. (3) now have superscripts, but this now is inconsistent with other ocurrences of $o_{1}$ and $o_{2}$ around Eq. (3)), and a careful revision will help readers to better understand the idea. For the experiments, it seems the baselines are not fine-tuned, and it's also not clear how the online algorithms are set-up for the offline setting. Providing details on how CSSC/PSE/CTRL is combined with CQL will also be helpful. In addition, comparison to existing offline algorithms will be helpful (e.g. refer to Offline Reinforcement Learning: Tutorial, Review, and Perspectives on Open Problems by Levine et al.).



**Summary Of The Paper:**

This paper proposes improving the generalization performance offline RL algorithms by using a new approach to aggregate observations based on the similarity of their expected future behavior.

**Summary Of The Review:**

The proposed approach seems novel, but the motivation and presentation of the idea is confusing. It is not clerar whether the reported performance improvement is due to the proposed aggregation method contributes, and further discussion or experiments with related works may be needed.

---

> ### Author Response · Authors · 2021-11-16
> **Response (Major comments)**
>
> We thank you for the important feedback regarding notational issues. We agree that the CTRL algorithm could be an interesting baseline to add in the setting. We are working on implementing CTRL on the offline Procgen case and will add results into a future revision. Below is a point-by-point response to your questions:
> 1. **Regarding POMDP vs MDP confusion:** We apologize for the unclear phrasing present throughout the paper. The setting of control from pixels is typically treated as an MDP for simplification of theory in works on Atari and standard MuJoco environments (half-cheetah, antmaze, etc), although MuJoCo requires frame stacking to be considered an MDP (see Zhang et al. 2019). In our case, the setting is a POMDP in that the true state of the system is hidden from the agent (RAM), and only high-dimensional observations are exposed to the agent. Confusion typically arises from the fact that classical POMDP solvers keep track of the belief state through some recurrence mechanism. In Procgen, however, such a mechanism was shown to not be mandatory but, for the sake of theoretical formulations, we treat the setup as a POMDP with a short history length. *We have updated, through the paper, any MDP notation to be in terms of POMDP, since this is the primary entity we consider in our derivations*.
>
> 2. **Regarding assumptions on $\mu$:** we do not make any specific distributional assumptions on $\mu$.
> However, for good performance, there are some desirable properties of $\mu$, which we will clarify here and in the updated text. Note that we are in the classical batch / offline RL setting with one important distinction: downstream task performance is evaluated on the entire distribution of levels / POMDPs. This means that in practice $\mu$ has to contain enough information about solving the training task, e.g. sufficient state-action coverage. In our experiments, $\mu$ was taken to be the PPO policy pre-trained on 25M frames of online Procgen (classical setup in literature, as is the case in Cobbe et al. 2020 and Raileanu et al. 2021), which was additionally corrupted with $\varepsilon$-greedy noise according to a linear schedule in order to improve the action coverage. The notions of representation learning and exploration are tightly coupled (see ”​​Model-free Representation Learning and Exploration in Low-rank MDPs” by Modi et al. 2021), and hence there needs to be some overlap with the optimal solution in the dataset.
>
> 3. **Regarding illustrative example:** The motivating example presented in the paper is actually a problem which agents face when learning on Procgen. Both levels are perceptually distinct due to different backgrounds, lengths of platforms, and agent’s position, and hence lead to dissimilar action sequences (as shown in Fig.1). On the other hand, the state values $V(o_t)$ for the first sequence of observations are 8.20, 8.05, 8.74, 9.06, 9.29, and 8.35, 8.30, 8.26, 8.55, 8.40 for the second sequence of states. We can see that values are much more similar in this coupling of tasks than actions and, if we group the states by their corresponding value magnitude (or quantiles), the encoder will learn to assign all states which look like the sequences above to a neighbouring latent representation, and hence, will allow better zero-shot generalization as it will learn to ignore backgrounds, platform length and small variations in agent’s position. *This information was also added into the main paper Sec.7.3 paragraph 2*. Additional information on similarity between pairs of $(o_1,o_2)$ and corresponding true latent states $(s_1,s_2)$ can also be found in Sec.7.3 paragraph 1.
>
> 4. **Regarding MDP vs POMDP experiments**: The experiments are performed on pixels, i.e. all agents are provided with an observation batch of dimensions $(n_{batch},H,64,64,3)$ where H is the number of timesteps (H=2 in our case). While some previous works call this setting “MDP”, our formulation relies on the distinction between observations and true latent states of the system, hence we treat it as a general POMDP. An alternative formulation could be to use the BlockMDP paradigm ("Provably efficient RL with Rich Observations via Latent State Decoding" by Du et al. 2019), but its identifiability assumption cannot reliably be verified in large domains, of which Procgen is one.
>
> 5. **Regarding DeepMDP and bisimulation:** Yes, you are right, this was a mis-formulation on our end (which has been fixed in the revision); DeepMDP indeed learns a set of metrics related to, but not exactly equal to bisimulation metrics.

---

> ### Author Response · Authors · 2021-11-16
> **Response (Minor comments)**
>
> 1. We re-used the notation that is more or less common within the RL community, in particular in “Knowledge representation for reinforcement learning using general value functions” by Comanici et al. (2019) and other works. If there is consensus among reviewers that the terminology is confusing, we will replace “cumulant” with another term, such as “auxiliary signal”.
>
> 2. You’re correct, in that POMDPs operate over belief states rather than states. In our case, the belief states are reduced to length one due to the existing ablations previously performed on history length by Cobbe et al. (2020) when proposing the Procgen benchmark. These ablations have shown that, although pixel-based control is typically partially observable, it is still possible to obtain a good policy by truncating the belief state to the most recent N observations (we use N=1). The paper “Improving sample efficiency in model-free reinforcement learning from images” by Yarats et al. (2019) explains the subtle distinction between POMDPs and MDPs for pixel observations in Section 3.3. The revision has been updated correspondingly.
>
> 3. Thank you for pointing out the notational issue with $\sup$ of vector-valued cumulants. In practice, we use a summary statistic of $c(o,a)$. If this is a vector - we use the norm of this vector ($\ell_1$ for the successor feature vector to approximate the state visitation density). We fixed the issue in the revision.
>
> 4. Please see our response to reviewer f74c.
>
> 5. This was a typo on our part, which is now fixed in the revision.
>
> 6. $c_{1,max}^\mu$ is an abuse of notation for $c_{max}^\mu$ for POMDP $M_1$, we apologize for this confusion.
>
> 7. Yes, you’re right - the idea is that randomness comes from $o_t$ having some distribution induced by the dataset $\mathcal{D}^\mu$. $G^\mu$ (or $G$, which is equivalent) is a deterministic function which transforms the distribution of $o_t$ into a potentially analytically intractable distribution, but for which we can compute empirical quantiles.
>
> 8. This is once again a typo. We corrected the issue in the revision.
>
> The remaining errors were typos and we apologize for having them in the first place - we also fixed them in the revision.

---

> ### Author Response · Authors · 2021-11-21
> **Additional results**
>
> We would like to follow up on your suggestion of comparing our GSF method against the CTRL algorithm. We have implemented Alg. 1 of CTRL in our codebase, with cluster length equal to 2 (due to the offline Procgen dataset structure being $(o,a,o’,r)$). We provide the results in Fig. 3 of the second revision. As it can be seen, CTRL+CQL performs a bit better than CQL, but still overall worse than GSF.

---

### Author Response · Authors · 2021-11-21
**Additional results**

We thank once again all reviewers for their constructive comments and hope we addressed all of them in the first revision. We are including the second revision of the paper, which has the CTRL algorithm (Mazoure et al. 2021) included as a baseline (see Fig. 3); moreover, CURL (Srinivas et al. 2020) was also included in revisions 1 and 2. We hope that these additional results help highlight the utility of GSF in the offline setting.

---

### Decision · Program_Chairs · 2022-01-20

**Decision:**

Reject

**Comment:**

The paper proposes a new offline RL technique to generalize across domains. The paper was initially confusing (i.e., MDP vs POMDP) and weak empirically.  The authors greatly improved the paper.  However, a the end of the day, it is still not clear why the proposed approach performs better than existing techniques.  We can think of the cumulant function with the discrete labels as essentially computing some statistics of future actions, observations and rewards.  This is what every self supervised technique does.  They differ in terms of their particular choice of statistics and architecture.  The paper does not sufficiently motivate the particular architecture.  Interestingly, in the experiments, the best statistics are cumulative rewards, which are closely related to the Q-values. In that case, it is even less clear why the approach should be beneficial since RL techniques that generalize across domains by learning state representations to predict Q-values seem very closely related.

Despite the updates to the paper, the POMDP references are still confusing.  The issue is that the paper embeds observations as if they were sufficient to predict future observations and rewards.  This corresponds to the memoryless approach where a policy is optimized based on the last observation instead of the history of past actions and observations.  Memoryless strategies are effective only when the last observation is a sufficient statistic, meaning that we really have a (near) fully observable MDP.  The paper should discuss this and acknowledge that the approach will suffer in domains where memory of past actions and observations is critical.